# Application of Machine Learning Algorithm on MEMS-Based Sensors for Determination of Helmet Wearing for Workplace Safety

**DOI:** 10.3390/mi12040449

**Published:** 2021-04-16

**Authors:** Yan Hao Tan, Agarwal Hitesh, King Ho Holden Li

**Affiliations:** 1School of Mechanical and Aerospace Engineering, Nanyang Technological University, Singapore 639798, Singapore; yh.tan@ntu.edu.sg; 2School of Computer Science and Engineering, Nanyang Technological University, Singapore 639798, Singapore; HITESH001@e.ntu.edu.sg

**Keywords:** helmet, appropriate use, microclimate, machine learning, logistic regression

## Abstract

Appropriate use of helmets as industrial personal protective gear is a long-standing challenge. The dilemma for any user wearing a helmet is thermal discomfort versus the chances of head injuries while not wearing it. Applying helmet microclimate psychrometry, we propose a logistic regression- (LR) based machine learning (ML) algorithm coupled with low-cost and readily available MEMS sensors to determine if a helmet was worn (W) or not worn (NW) by a human user. Experiment runs involving human subject (S) and mannequin experiment control (C) groups were conducted across no mask (NM) and mask (M) conditions. Only ambient-microclimate humidity difference (AMHD) was a feasible parameter for helmet wearing determination with 71 to 85% goodness of fit, 72 to 76% efficacy, and distinction from control group. Ambient-microclimate humidity difference’s rate of change (AMHDROC) had high correlation to helmet wearing and removal initiations and was quantitatively better in all measures. However, its feasibility was doubtful for continuous use beyond 1 min due to plateauing AMHD response. Experiments with control groups and temperature measurement showed invariant response to helmet worn or not worn with goodness of fit and efficacy consolidation to 50%. Results showed the algorithm can make helmet-wearing determinations with combination of analysis and use of data that was individually authentic and non-identifiable. This is an improvement as compared to state of the art skin-contact mechanisms and image analytics methods in enabling safety enhancements through data-driven worker safety ownership.

## 1. Introduction

Appropriate use of the helmet as safety protection against workplace related injuries and deaths has long been a complex problem riddled with trite and questionable solutions. Here, the term appropriate use distinguishes itself from the naïve compliance safety approach in having workers to wear helmet at all times. Wearing helmet at all times causes thermal discomfort with heat and humidity building up on the user’s head [1], affecting the user’s mental state and physical performance in the process. This is more exacerbated in tropical countries like Singapore and its South East Asian neighbours, where experienced temperature can be over 30 degrees Celsius with humidity ranging from 60 to 90% RH. Research thus far [1,2,3,4,5,6,7] have focused on various cooling strategies and studies that measured users’ thermal comfort and their work performance.

Anecdotal factors in workplace safety and health (WSH) play a significant role on top of traditionally attractive approaches towards thermal comfort and human performance. In hot and humid environments, an industrial safety helmet has to compete against masks and balaclavas for head space. Sacrificing a mask for a helmet would be easy for safety compliance, but it obscures a worker’s vision as perspiration can enter his eyes with little barrier. It can also increase his risk of skin-related occupational disease due to increased direct sunlight exposure for prolonged durations. Conversely, sacrificing a helmet for a mask would not only risk non-compliance and statutory penalties, but it also loses mechanical impact protection that a helmet’s hard shell provides. Singapore, as part of its forward-looking efforts, is targeting to implement best WSH practices by 2028 [8] with strengthening safety ownership and technology-enabled WSH as two thirds of its overall strategy. Part of safety ownership aims at facilitating worker involvement via safety management, worker education, and unsafe work condition reporting systems as avenues to achieve customized, last-mile WSH measures created by persons who have direct interest in preserving themselves. However, evidence of such safety dilemmas remains hidden when considered alongside workplace goals of remaining profitable despite a minority of flouters, which effectively causes genuine work redesign suggestions or unsafe work grievances to go undetected. This is where technology-enabled WSH can fill the gap with data-driven research and innovation.

Our aim is to determine whether a helmet worn by a human subject using machine learning (ML) algorithm(s) with low-cost and readily available MEMS sensors can leverage individually authentic and non-identifiable helmet sensory information as a foundation for a data-driven worker safety ownership solutions. Although helmet researchers remain split on various thermal comfort interventions, helmet microclimate psychrometric response to helmet wearing consistently shows two distinct states between ambient and helmet microclimate parameters [1,3,7] due to metabolism and perspiration dissipation via a human wearer’s head. This provides an authentic and strong physiologically-based phenomenon for helmet wearing determination [9,10,11,12,13,14,15,16,17]. Besides psychrometry, other helmet safety research focused on enabling new levels of safety advisories by adding environmental hazard sensors with communication devices to sense for increasing danger [18,19,20] or brain-computer-interface with physiological sensors for emotion detection [21]. Furthermore, application of ML in small-scale applications is a new development as recent works explore fracture mechanics and additive manufacturing to overcome analytical and empirical limitations when solving complex engineering problems [22,23].

Running ML algorithms with MEMS sensors and embedded computing platforms for small-scale applications comes with the advantage of overcoming false responses created by universally-prescribed fixed-decision thresholds failing to deal with localized deviations. Instant ML modelling and predictions of immediate MEMS sensor measurements are therefore well-positioned to facilitate calibrated decision thresholds’ creation and decision making based on immediate local ground truths. At the same time, the solution must also be cost effective for eventual commercial acceptance and be of appropriate form factor for user acceptance. This is in contrast to recent technologies such as image analytics [24] which requires a large number of cameras and direct view of sight which are impractical in many worksites. Inherently, these bodies of works point to a research gap of using binary output ML algorithm with ambient-microclimate difference (AMD) as input in characterizing not worn (NW) and worn (W) states. If proven feasible, data logged will not only detect helmet misuse but also suggest if a helmet may have been over-prescribed for work, thus providing an unbiased and reliable database in guiding work redesign efforts towards appropriate use.

The objective of this algorithm assessment study was to explore ML’s logistic regression (LR) modelling in helmet wearing determination by analyzing goodness of fit and determination efficacies between human subject (S) and mannequin experiment control (C) groups with respect to their ground truths. Two flouting scenarios were considered. First scenario was represented by C group where a non–human object wears a helmet’s microclimate at all times. The second scenario was represented by assembling a subject–control (SC) group where a helmet was worn by a non–human object after proper calibration. No mask (NM) and mask (M) runs were done for analysis under mask or balaclava conditions. Four AMD psychrometric parameters were studied for feasibility; each parameter was used independently in creating its LR model. They were as follows: (1) ambient-microclimate temperature difference (AMTD); (2) ambient-microclimate humidity difference (AMHD); (3) ambient–microclimate temperature difference’s rate of change (AMTDROC); and (4) ambient-microclimate humidity difference’s rate of change (AMHDROC). By our definition, rate of change is a parameter’s derivative with respect to time.

Section 2 of this paper presents the methods and materials, detailing algorithm, prototype, data collection scope, and experiment procedures. Section 3 covers results and analysis. Section 4 concludes the study, highlighting significance to the industrial safety helmet and safety landscape.

## 2. Methods and Materials

### 2.1. Helmet Microclimate Machine Learning Algorithm

Our proposed algorithm (Figure 1) uses a 2 min calibration phase to classify helmet NW and W states into an immediate use LR ML model. Calibration phase duration was based on statistically driven data sufficiency for each binary state. Using central limit theorem (N ≥ 30), each binary state was to contain minimum 30 datapoints, creating calibration requirement of at least 60 datapoints. Considering data refresh rate of low-cost and readily-available MEMS sensors were 2 s and ideal calibration time should be as short as possible, this resulted in a total calibration duration of 2 min which was practicable for a worker who started making work preparations and was not too long as to impede work. During calibration, the helmet started with NW state for the first minute followed by W state in next minute, forming a data baseline for LR model calibration. Once calibrated, the LR model was deployed for continuous monitoring. In continuous monitoring phase, psychrometric measurements were taken continuously with each new measurement acting as calibrated LR model’s input in determining output to be either 1 representing W state or 0 representing NW state.

In this study, four AMD psychrometric-based parameters were independently used to generate their respective LR model. AMTD and AMHD were calculated by taking the difference between ambient and microclimate sensor measurements. AMTDROC and AMHDROC were calculated by subtracting respective AMTD and AMHD value from its immediately preceding value; this difference was then divided by sensor’s refresh rate of 2 s. To implement this algorithm, LR ML used was from sci-kit-learn python ML library with cross-fold validation (CV) set at 30 to randomize calibration data for a statistically significant number of times in searching for the best LR model possible.

### 2.2. Helmet Dataloggers

The experimental setup utilized a data-logging helmet attachment with replaceable MEMS sensors. To demonstrate preservation of helmet hard shell’s mechanical protection, Future Assault Shell Technology (FAST) styled helmets with built-in velcro pads were used to secure these attachments, as shown in Figure 2. Two identical helmets were used to facilitate experiment proceedings. Each data logger was assembled with a 10,000 mAh 5 V USB powerbank battery and an Onion Omega 2 Plus single-board-computer (Onion Corporation, Boston, MA, USA, average power use of 0.6 W), accompanied by Arduino–microcontroller docker accessory (estimated average power use of 0.6 W) connected to two DHT–22 temperature and humidity sensors (Aosong Electronics Co Ltd., Guangzhou, China, average power use of 0.0033 W per sensor), as shown in Figure 3. One sensor was placed within the helmet’s core to measure microclimate temperature (MT) and microclimate-relative humidity (MH) while the other was placed at the helmet’s external surface to measure ambient temperature (AT) and ambient-relative humidity (AH). Continuous use starting with full battery charge was estimated at 41 h based on power specification and similar prototypes having demonstrated 36 h of operation. To eliminate erroneously repeated readings from influencing data’s statistical significance, data-loggers were synchronized to record at DHT–22 sensor’s 2 s measurement refresh rate.

### 2.3. Experiment Proceedings

Two sets of 30 experiment runs were conducted for data collection. First set was under NM condition while second was M condition. These runs were conducted in room conditions and across non-consecutive days to ensure reproducibility under varying conditions in an indoor setting, as shown in Figure 4.

Each run was an action sequence designed to contain ground truth information from calibration followed by helmet wearing and removals during continuous monitoring. Ground truths, by common definition, are information from direct observation. Therefore, in this study, ground truth was a helmet’s actual state of either being worn or not captured via camera recording. Each action sequence consisted of a 2 min calibration period (first minute NW, second minute W). This was followed by a 4 min test period by alternating the helmet between 1 min NW and W states. Both S and C were subjected to the same abovementioned sequence simultaneously.

A head mannequin was used as experimental control to simulate reasonably sophisticated means to outsmart the helmet. Two flouting scenarios were then analyzed. First, a total abandonment scenario of using a non-human substitute at all times was utilized and was represented by data from C group. Second, a misuse scenario was utilized where the helmet was properly calibrated and then transferred to a non-human substitute in attempt to outsmart the algorithm. This was represented by combining each run’s S group calibration data with its C group continuous monitoring data; this assembled data group was thereby referred as SC group.

## 3. Results and Discussion

This section presents results from direct data observations following with analysis on algorithm’s goodness of fit, efficacy, and security for discussion. For visualization, an experiment run each from NM and M conditions were randomly selected to represent respective datasets. Random sampling showed similar trends across all runs. Analysis would primarily focus on humidity parameters as temperature response was invariant in Figure 5 and Figure 6. Visualization of results was done using Excel standard plotter tools and analysis was made using Python’s sci-kit-learn ML library. Unless specified, these findings cover both NM and M conditions.

### 3.1. Direct Observations from Raw Data Logs

From direct observations, microclimate humidity (MH) was the only significant responsive parameter for helmet wearing determination. This was observable in Figure 7 and Figure 8 as exponential growth and decaying MH waveforms in response to helmet wearing and removing, respectively. Microclimate temperature (MT) was instead invariant. However, MH alone could present a single point failure where sensor error would be extremely hard to detect and hence AMHD was needed to realize MH utilities with ambient-relative humidity (AH).

AMHD preserved AH’s characteristics as a responsive and authentic parameter suited for helmet wearing determination. This was possible as AH remained relatively constant, resulting in exponential-like growth and decay AMHD waveforms in Figure 9 and Figure 10. This meant that AMHD provided a means to overcome MH’s single point failure weakness while maintaining binary state calibration data needed for LR modelling.

Ambient-microclimate humidity difference’s rate of change (AMHDROC), in Figure 11 and Figure 12, consequently showed impulse-like response to helmet wearing and removals. This impulse response was directional, as evident by an immediate polarity changeover in AMHDROC values when helmet was worn or removed. It also tended to zero as AMHD plateaued towards the end of the1 min wearing and removal sequences which meant AMHDROC could not sustain a response for longer durations.

MH and its related AMD parameters also provided authentication capabilities in identifying a human wearer. This was in part due to invariant response exhibited by experimental control in MH-C group data in Figure 7 and Figure 8. When compared to MH-S group’s exponential-like growth and decay, this meant that MH response was unique to a human wearer.

### 3.2. Calibration (First 2 min) Data’s Goodness of Fit as Logistic Regression Model

Goodness of fit served as metric in quantifying how well datapoints are associated as a calibrated LR function. In an ideal situation, a 100% goodness of fit represents a perfect alignment between datapoints and modelled function. An example is shown in Figure 13 with a LR model being modelled based on AMHD data, and in Figure 9 with NW datapoints associated to Helmet State = 0 and W datapoints to Helmet State = 1. Outlier points between −7 to −6%RH AMHD then contributed to a goodness of fit spread between 71 to 85%, leftmost of Figure 14. Invariant parameters only demonstrated overall consolidation to 50% goodness of fit, with full results as shown in Figure 14.

AMHD was the best parameter in goodness of fit even though it was quantitatively second, with AMHD-S ranging from 71 to 85%, leftmost in Figure 14. On the other hand, AMHD-C group showed a considerably large range of 50 to 92% for NM condition and consolidated at 50% for M condition. For AMHD, a clear distinction was observable between AMHD-S and AMHD-C groups to facilitate identification of flouting attempts. AMHDROC was the quantitatively best parameter with AMHDROC-S ranging from 82 to 92%. Most of the time, AMHDROC calibration data were zeroes and therefore produced near-vertical LR decision thresholds at zero. This exceptional goodness of fit was due to small but quantitatively favorable deviations in AMHDROC values during calibration phase. There was also a clear distinction between AMHDROC-S and AMHDROC-C groups, with the latter consolidated at 50%. Even though AMHDROC was quantitatively superior and performed similarly, it lacked efficacy in helmet wearing correlation for extended periods that AMHD possessed. This will be further discussed in Section 3.4.

Most control groups can be easily picked out via a 50% goodness of fit. Experiment controls exhibited a consolidation at 50% goodness of fit caused by vertically arranged calibration datapoints that were quantitatively unfavorable to LR modelling. This vertical arrangement was consequent of same value datapoints being associated to both NW and W helmet states. Exceptions often occurred under NM condition which showed wide spreads from 50% to 92%, and also due to small deviations in parameter value. Therefore, it can be reasonably inferred with a 50% goodness of fit that a non-human wearer was present.

### 3.3. Efficacy, LR Model’s Ability for Continuous Monitoring

Efficacy was defined as LR model’s ability to make correct determination during continuous monitoring. C group efficacy also encompassed our first flouting scenario of non-compliant calibration and continuous helmet use. Ideally, a 100% efficacy would represent all outputs as perfect reflection of ground truth. For analysis, continuous data were processed together with each corresponding LR model. Each datapoint served as input for the LR model to generate an output result. This output was compared against its corresponding moment in the ground truth video, thus creating a series of comparisons for each run. These series of comparison were normalized on a 0 to 100% scale, with full results as shown in Figure 15.

AMHD was the best parameter in terms of efficacy even though it ranked second quantitatively, with AMHD-S scoring between 72 to 76%, as shown on the leftmost column in Figure 15. This meant that determinations were correct most of the time. Moreover, such efficacies were distinct from AMHD-C group which showed a range of 48 to 58%. Hence, AMHD was a clear distinction between S and C groups indicating that non-compliant calibration can be identified during continuous monitoring when false determinations occur about half the time. Similarly to be discussed in Section 3.4, AMHDROC was quantitatively better with AMHDROC-S ranging from 88 to 95% efficacy; AMHDROC-C group continued to consolidate at 50% with clear distinction between its S and C groups.

### 3.4. Security, LR Model’s Ability to Quantify Flouting Attempts

Security was defined as LR model’s ability to quantify a non-human wearer after proper calibration. This encompassed our second flouting scenario of abandoning a helmet after adhering to calibration procedures. Ideally, security can be inferred via distinct efficacies between a parameter’s S and C group; the greater the better, as similar efficacies would be unfathomable. For analysis, a subject-control (SC) group was created by cross assembling a run’s S group calibration data with C group continuous monitoring data thus creating inputs from a non-human source to a human-calibrated LR model. Similarly, SC group LR model would create a series of comparison to be normalized on a 0 to 100% scale. The full results comparing S with SC groups are shown in Figure 16.

AMHD was the best parameter in security even though it was quantitatively ranked second after AMHDROC. This observed distinction was due to incompatibility between S group generated LR decision threshold and C group continuous monitoring data. To elaborate this, AMHD-S calibration data in Figure 7 (top) showed that a −6%RH at the start of calibration gave rise to a −6% RH LR model decision threshold in Figure 13. However, when AMHD-C continuous monitoring values between 4 to 5%RH in Figure 7 (bottom) were used as inputs, the LR modelling output would always be a 1. As a result, outputs were syntactically correct for half the time resulting in a 50% efficacy for SC. Given that C group data was invariant, the same superficial 50% efficacy would be produced even if AMHD-C values were 0 outputs. This would apply similarly to M condition in Figure 10, with a −12%RH in this instance.

Even though SC efficacies demonstrated a tighter consolidation, both C and SC groups demonstrated 50% efficacy consolidation across Figure 15 and Figure 16. This would enable quantitative inference of a flouting event when a 50% efficacy was presented. AMHDROC also showed a distinction between S and SC groups, but its consistent quantitative superiority was solely rooted in the study’s 1 min helmet wearing and removal time period. For continous monitoring beyond 1 min, AMHDROC values will lose correlation to helmet W state, as AMHD plateau was prefaced in Section 3.1. Poorly correlated AMHDROC input data then resurfaces incompatbility issues between LR decision threshold with continuous monitoring measurements. Corresponding drops in AMHDROC-S goodness of fit and efficacy would occur if either wear (W) calibration phase or continuous use was extended beyond 1 min. Therefore, AMHDROC could not be relied upon as a continuous monitoring parameter.

## 4. Conclusions

A machine learning LR algorithm was studied for helmet wearing determination efficacy. To assess if a helmet was being worn by a human subject across location, time, and environments, the proposed algorithm consisted of a 2 min immediate calibration phase and a continuous monitoring phase. Four psychrometric parameters, AMTD, AMHD, AMTDROC, and AMHDROC, were studied due to consistent binary state psychrometric observations in past helmet research. Prototype helmets were built using low-cost and readily available MEMS sensors to demonstrate current technology feasibility and algorithm experimentation. S and C groups were simultaenously tested through an action sequence for a statistical significant number of runs and across NM and M conditions. Post-analysis was done for the data collected. Temperature was observed to be an invariant or slow response to helmet wearing and thus analysis focused primarily on humidity related parameters.

AMHD was found to be the best parameter for helmet wearing determination. When worn by a human subject, AMHD calibration data demonstrated 75% goodness of fit as an LR model which meant data was largely accounted for. When tested on an experimental control mannequin head, AMHD instead demonstrated highly concentrated goodness of fit at 50% or huge spreads. This means it would be possible to infer a human from a non-human wearer when presented with an LR model’s goodness of fit. This distinction was carried over to helmet determination efficacy analysis which was also the first flouter scenario where AMHD-S group demonstrated 70% efficacy in making correct determinations, whereas efficacy was distinctively lower at 50% in C group. This meant that during continuous monitoring, a properly calibrated helmet would perform as intended most of the time while non-complying helmet calibration would often give false alarms. AMHD was again useful in detecting second flouter scenario after calibration procedures were adhered to. This was due to fundamental response differences between S and C groups observed during calibration that resulted in incompatiblity between LR decision thresholds and continuous monitoring input data. As consequence of model and measurement incompatibility, efficiacy in SC group dropped to 50% when using C group continuous monitoring data to predict helmet state through S group calibration data-created LR models. In practice, this means AMHD could still provide countermeasures through false alarms when a flouter tries to outsmart the solution after proper calibration. In culmination, a combination of AMHD goodness of fit and efficacy analysis was sufficient to confidently determine if a helmet was worn by a human or non-human wearer.

AMHDROC was instead found feasible in identifying specific instances of helmet wearing and removal events due to its impulse-like directional responses. In terms of helmet wearing determination, the parameter was quantitatively better than AMHD across goodness of fit, efficacy, and security analysis but carried an inherent flaw. This quantiative advantage was only due to a 1 min helmet wearing time period designed into experimental procedures which gave AMHDROC values a high correlation to helmet wearing and removal events. In practice, AMHDROC correlation would diminish significantly as AMHD pleteaued beyond 1 min of continous use. AMHDROC LR decision thresholds and continuous measurements would then be incompatible resulting in complete determination failure. When considered holistically, AMHDROC could be best used as a supporting parameter in automating ground truth associations during calibration phase.

These first findings presented in this paper are important to small-scale ML MEMS research in enabling forefront WSH practices for tropical workplaces. As a proof of concept shown in this work, helmet psychrometric datalogs can be analysed to confirm helmet worn status, providing authentic personalization without creating privacy issues related to other wearable solutions. This non-identifiable data collection takes care of any potential conflict in personal privacy issues, and it is in line with Singapore’s effort with worker safety ownership. There is high potential for eventual commercial application through the use of two low-cost and readily available MEMS sensors, a single parameter, non–identifiable psychrometric data, and low-power computing device with local ML capabilities for helmet wearing determination. Furthermore, its form factor as a non-contact helmet wearable attachment circumvents extreme discomfort issues that current skin-contact wearable instrumentations introduce in tropical workplaces. Besides tropical Singapore, this study has potential to be tested in other climates for further research.

## Figures and Tables

**Figure 1 micromachines-12-00449-f001:**
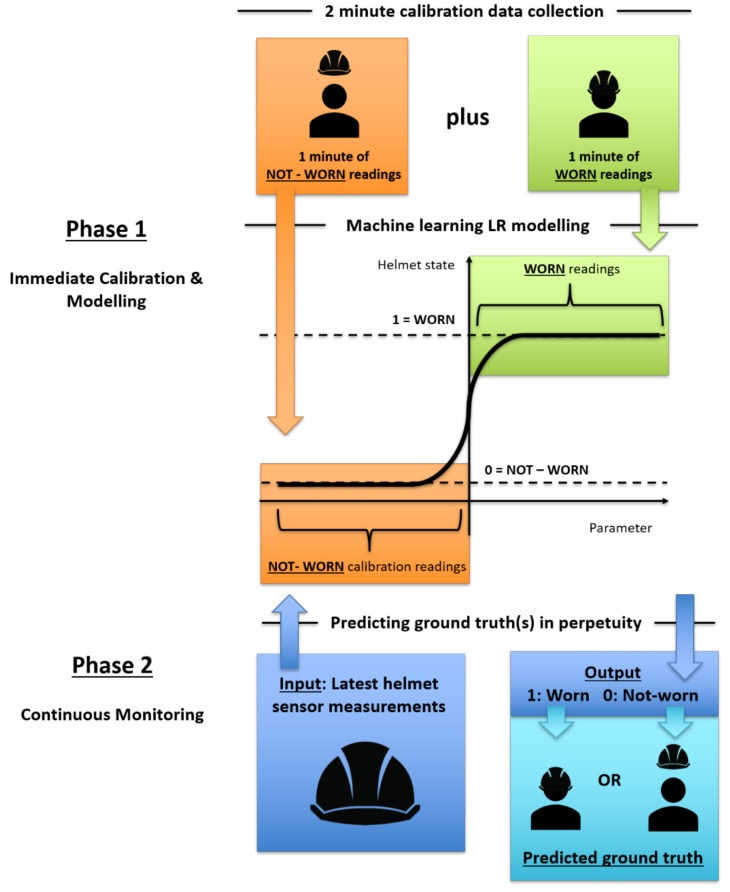
Schematic of machine learning logistic algorithm for helmet wearing determination.

**Figure 2 micromachines-12-00449-f002:**
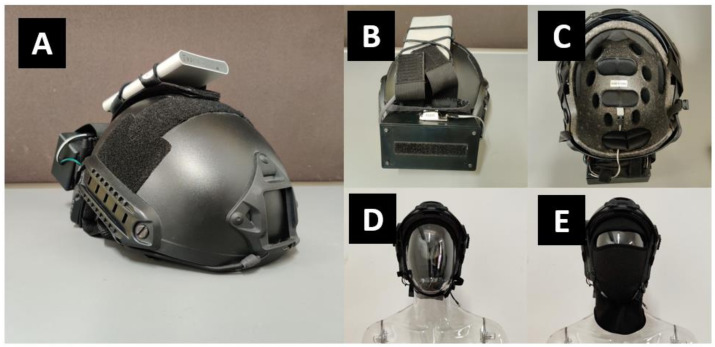
Photos of prototype of Future Assault Shell Technology (FAST) style helmet. (**A**) Isometric view; (**B**) rear view with datalogger and ambient sensor; (**C**) internal view with microclimate sensor location; (**D**) prototype on mannequin head without a mask; (**E**) prototype on mannequin head with a balaclava mask.

**Figure 3 micromachines-12-00449-f003:**
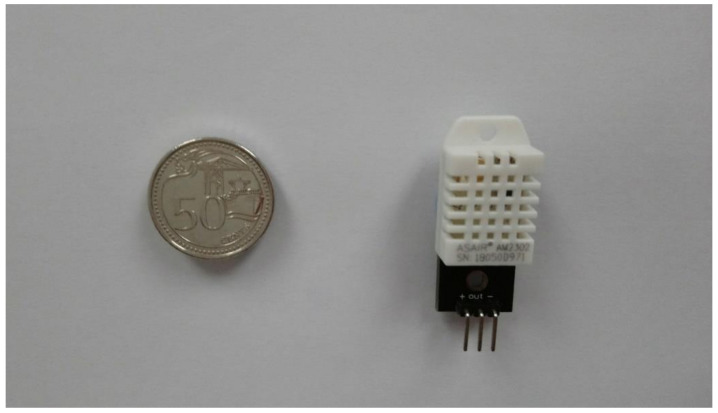
DHT–22 Humidity and Temperature sensor (Aosong Electronics Co Ltd., Guangzhou, China) with breakout board. Singapore 50 cent coin for scaling.

**Figure 4 micromachines-12-00449-f004:**
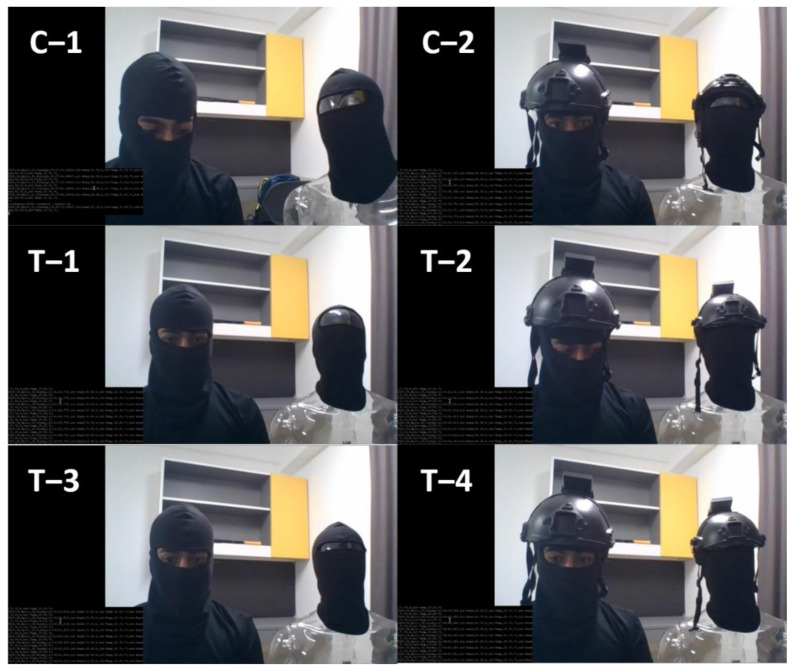
Action sequence of a data-collection run with balaclava mask on human subject and mannequin experiment control. C–1 and C–2 represents NW and W states during calibration phase respectively. T–1 through 4 represents 4 min test period.

**Figure 5 micromachines-12-00449-f005:**
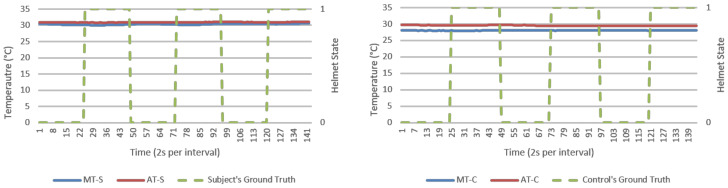
Temperature (°C) versus time (2 s per interval) data logs from a typical no mask (NM) run, human subject (**left**), and experiment control (**right**). Helmet state 0 = not worn (NW) and 1 = worn (W).

**Figure 6 micromachines-12-00449-f006:**
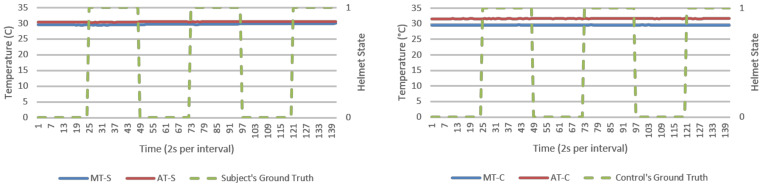
Temperature (°C) versus time (2 s per interval) data logs from a typical M run, human subject (**left**), and experiment control (**right**). Helmet state 0 = NW and 1 = W.

**Figure 7 micromachines-12-00449-f007:**
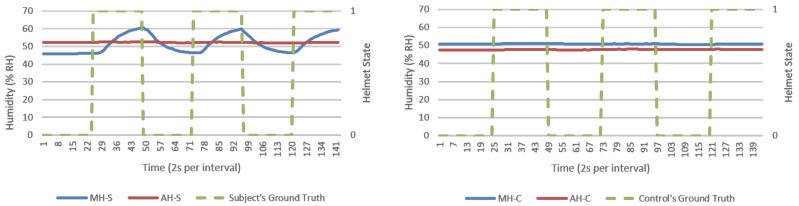
Relative humidity (% RH) versus time (2 s per interval) data logs from a typical NM run, human subject (**left**), and experiment control (**right**). Helmet state 0 = NW and 1 = W.

**Figure 8 micromachines-12-00449-f008:**
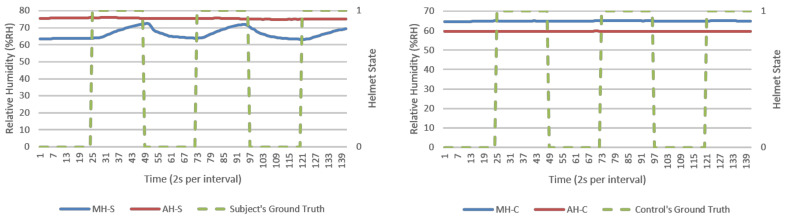
Relative humidity (%RH) versus time (2 s per interval) data logs from a typical M run, human subject (**left**), and experiment control (**right**). Helmet state 0 = NW and 1 = W.

**Figure 9 micromachines-12-00449-f009:**
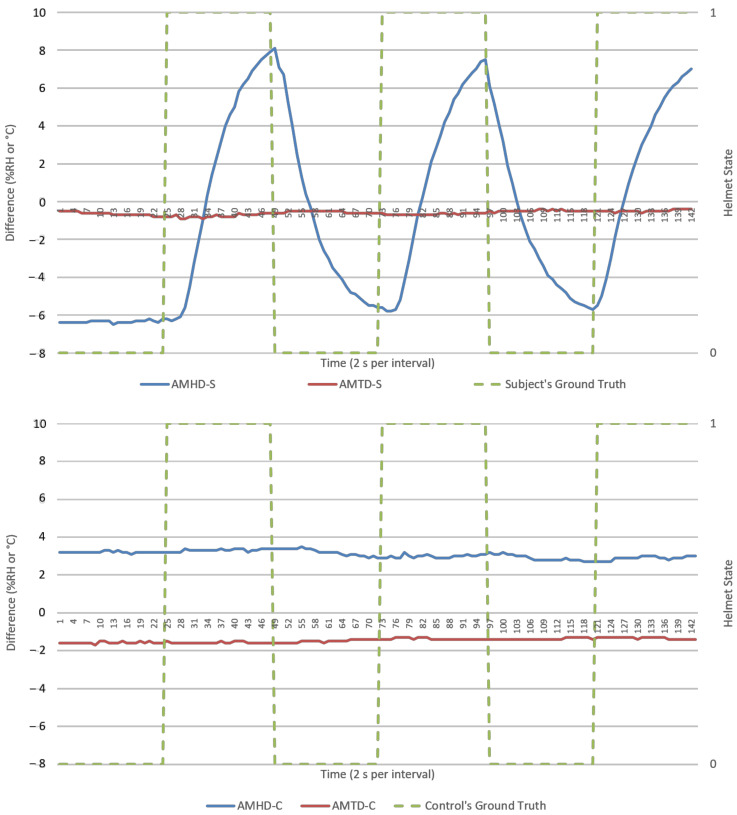
Ambient-microclimate differences (AMDs) versus time (2 s per interval) from a typical NM run, human subject (**top**), and experiment control (**bottom**). Helmet state 0 = NW and 1 = W.

**Figure 10 micromachines-12-00449-f010:**
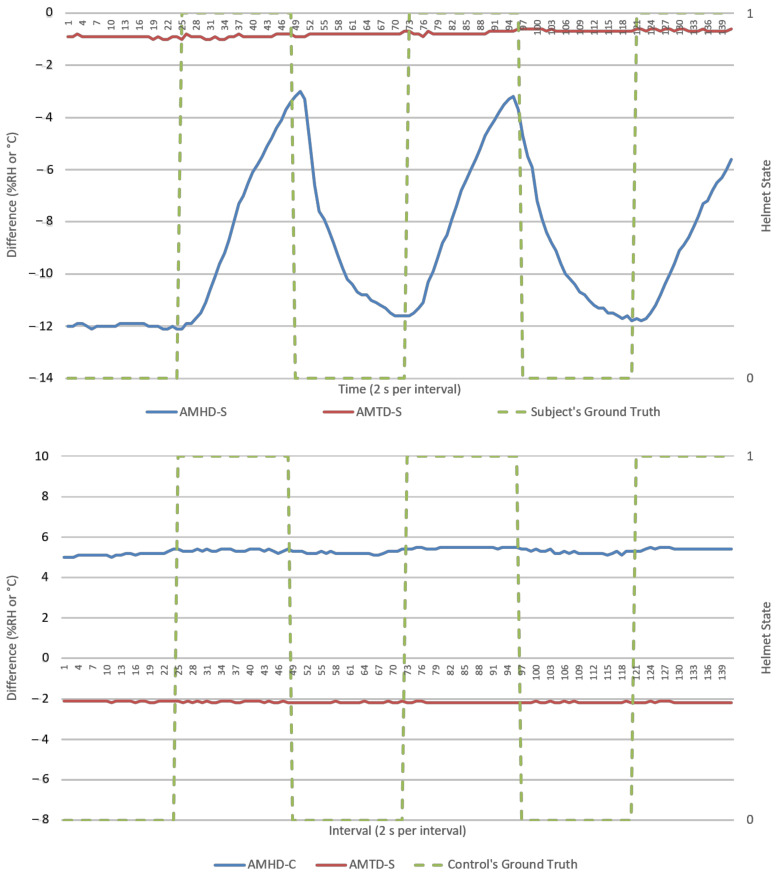
AMDs versus time (2 s per interval) from a typical M run, human subject (**top**), and experiment control (**bottom**). Helmet state 0 = NW and 1 = W.

**Figure 11 micromachines-12-00449-f011:**
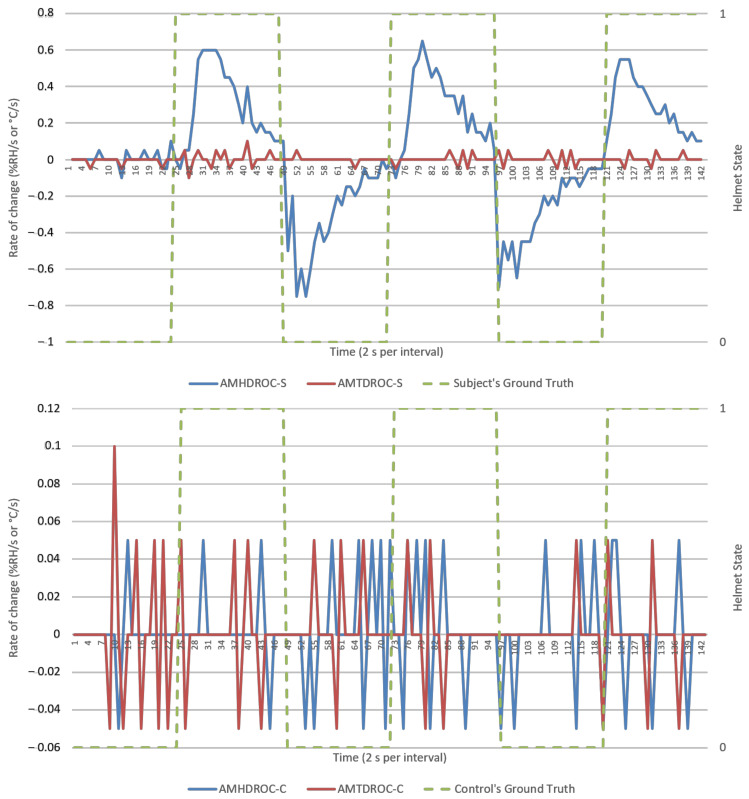
Ambient-microclimate difference’s rate of change (AMDROCs) versus time (2 s per interval) from a typical NM run, human subject (**top**), and experiment control (**bottom**).

**Figure 12 micromachines-12-00449-f012:**
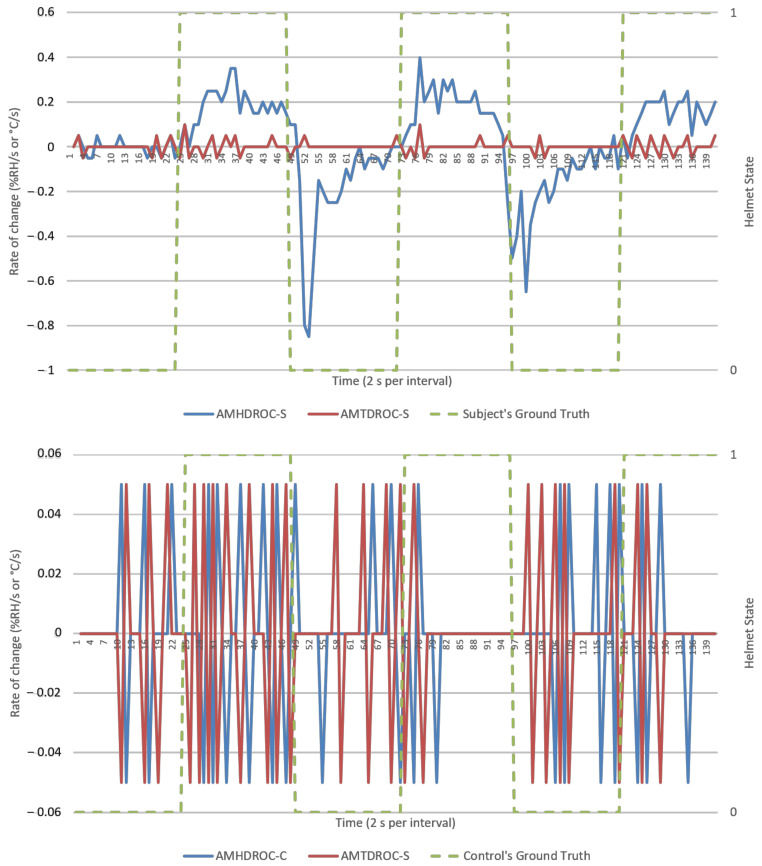
AMDROCs versus time (2 s per interval) from a typical M run, human subject (**top**), and experiment control (**bottom**). Helmet state 0 = NW and 1 = W.

**Figure 13 micromachines-12-00449-f013:**
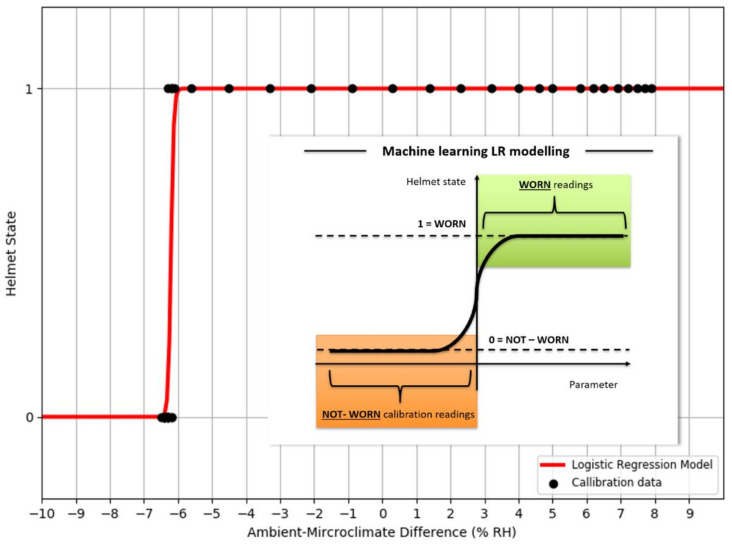
A typical ambient-microclimate humidity difference (AMHD) calibrated logistic regression model from a human subject under NM condition. Red line represents resulting LR function with decision threshold of −6% RH. Black dots represent NW datapoints associated to Helmet State = 0 and W to Helmet State = 1. The insert is the machine learning LR model as shown in Figure 1.

**Figure 14 micromachines-12-00449-f014:**
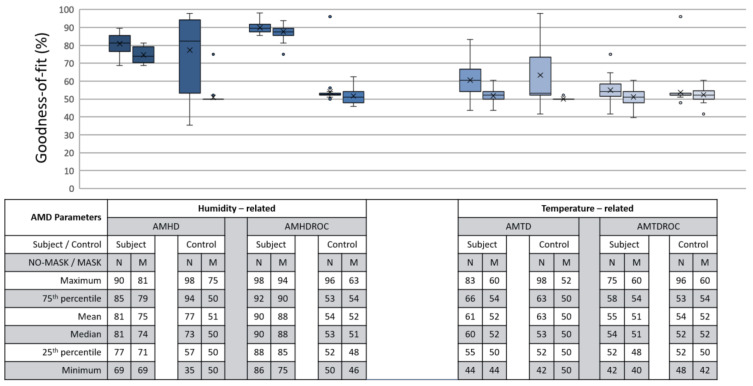
Goodness of fit distribution between subject (S) and control (C) groups in LR modelling represented in percentages. Each box contains 30 respective entries. AMHD (leftmost) has a mean of ~80% under N condition and ~75% under M condition.

**Figure 15 micromachines-12-00449-f015:**
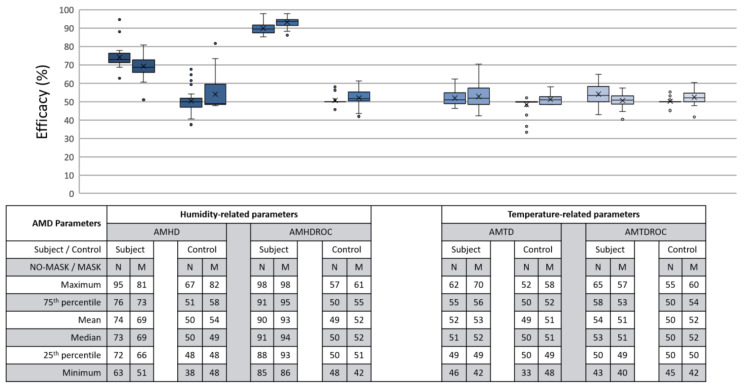
Efficacy between S and C groups during continuous monitoring. Each box contains 30 respective entries. AMHD (leftmost) has a mean of 74% under N condition and 69% under M condition.

**Figure 16 micromachines-12-00449-f016:**
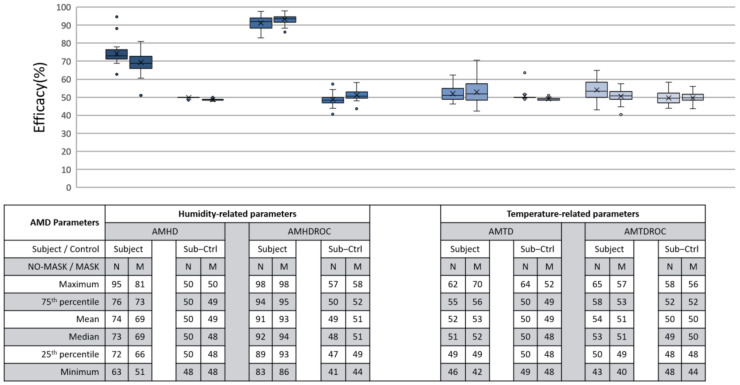
Efficacy between subject (S) and subject-control (SC) groups during continuous monitoring. Each box contains 30 respective entries. AMHD (leftmost) demonstrated clear distinction between S and SC groups.

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
