# Peer review of "Application of Machine Learning Algorithm on MEMS-Based Sensors for Determination of Helmet Wearing for Workplace Safety"

_micromachines, 2021, doi:10.3390/mi12040449_

Round 1
Reviewer 1 Report
Authors proposed interesting helmet wearing techniques for workplace safety with temperature and humidity MEMS sensors and applied the sensors to correct the positions of the helmet. Using machine learning algorithms, the logistic regression could be effective to reduce thermal discomfort. This method could be useful to avoid the accidents in the dangerous workplace. I never see this kind of the research to find the right positions of the MEMS sensors and to correct the helmet wearing methods. The measured temperature and humidity results looks reasonable. There are no problems of the motivation, research methods, and measured results in the manuscript.
However, there are some grammar mistakes so it is better to ask your colleague faculty to check the English grammar or use professional English services in entire manuscript because some units and values need to have some spaces and countless words cannot use "the". Literature search information is very limited. Therefore, authors had better provide more previous research papers. Therefore, the manuscript needs to be revised.
1. Authors need to use abbreviated journal names.
2. Authors need to provide abbreviated conference name in reference.
3. Authors need to provide the city and country and date information in Ref. [14].
4. Please remove the space in Line 399. Please check others.
5. Please increase label sizes in Figures 14, 15, and 16.
6. Figure 13 labels are not clear to be seen.
7. Two Figures 12 are too small so it is better put those Figures in the top and bottom positions.
8. In Figure 11 (a), I am wondering why relative humidity temperature are varied.
9. In the introduction sections, authors need to show the problems of the previous research for helmet research. If possible, please provide the machine learning methods related to helmet research.
10. In Line 255, why authors select only 2 minutes ?
11. Please use dash and minus symbols properly.
12. Data availability section is missing.
Author Response
The authors are deeply appreciate for the invaluable comment by the reviewers. Attached is the point by point response to the comments.
Moreover, the paper has been proof read by a professional for English language and grammar.
Thank you once again.

Reviewer 2 Report
Interesting work and use of ML algorithms.
The authors need to address the following:
- How did the authors choose the amount of data they used is sufficient?
- Some plots (i.e. FIg 13) need to be improved.
- Recent works of using ML for small scale applications need to be cited (e.g., X. Liu et al., Acta Materialia, 2020 and L. Lu et al., PNAS, 2020)
Author Response
The authors are deeply appreciate for the invaluable comment by the reviewers. Attached is the point by point response to the comments.
Thank you once again.

Round 2
Reviewer 1 Report
Authors provided revision quite well so the manuscript can be published.